# Efficacy and Safety of Omija (*Schisandra chinensis*) Extract Mixture on the Improvement of Hyperglycemia: A Randomized, Double-Blind, and Placebo-Controlled Clinical Trial

**DOI:** 10.3390/nu14153159

**Published:** 2022-07-30

**Authors:** Da-Som Kim, Hyang-Im Baek, Ki-Chan Ha, Youn-Soo Cha, Soo-Jung Park

**Affiliations:** 1Healthcare Claims & Management Incorporation, Jeonju 54858, Korea; dasomki07@gmail.com (D.-S.K.); hyangim100@woosuk.ac.kr (H.-I.B.); omphalos9121@hanmail.net (K.-C.H.); 2Departments of Food Science & Human Nutrition, Jeonbuk National University, Jeonju 54896, Korea; cha8@jbnu.ac.kr; 3Department of Food Science & Nutrition, Woosuk University, Wanju 55338, Korea; 4Korea Food Research Institute, 245, Wanju 55365, Korea; 5Department of Sasang Constitutional Medicine, College of Korean Medicine, Woosuk University, Jeonju 55338, Korea

**Keywords:** Omija, *Schisandra chinensis*, soybean, Omija extract mixture, glycemic control, type 2 diabetes, clinical trial

## Abstract

A previous animal study demonstrated that the administration of Omija extract and soybean mixture (OSM) improved glycemic control in the type 2 diabetes model. In this study, we conducted a 12-week, randomized, double-blinded, and placebo-controlled clinical trial to determine the effects of OSM in humans with hyperglycemia. Participants with fasting plasma concentrations of 100–140 mg/dL were enrolled (*n* = 80) and administered either OSM or placebo products for 12 weeks. The outcomes included measurements of efficacy (fasting plasma glucose (FPG), postprandial glucose (PPG), fasting plasma insulin (FPI), postprandial insulin (PPI), hemoglobin A1c (HbA1c), C-peptide, fructosamine, and lipid parameters) and safety at baseline and at 12 weeks. After the intervention, the OSM group showed significantly decreased levels of FPG, PPG (30, 60 min), PPI (60 min), insulin area under the curve (AUC), fructosamine, and low-density-lipoprotein (LDL) cholesterol compared to the placebo group. No clinically significant changes in any safety parameter were observed. Therefore, it is hypothesized that OSM supplementation is an effective and safe functional food supplement for humans with hyperglycemia.

## 1. Introduction

According to the WHO, at least 171 million people worldwide suffer from diabetes, which is likely to be more than doubled by 2030, thereby reaching 366 million. Furthermore, type 2 diabetes mellitus (T2DM) accounts for 90–95% of diabetes cases [1,2,3].

Hyperglycemia is a representative symptom in T2DM patients, manifested by the rapid rise in plasma glucose levels due the pancreatic α-amylase hydrolyzed starch and absorption of glucose in the small intestine by α -glucosidases [4]. It has been connected to the beginning of diabetic complications in T2DM patients and leads to the generation of free radicals and oxidation-related damage in the retina, renal glomerulus, and peripheral nerves [5,6].

T2DM is a major risk factor for cardiovascular morbidity and mortality. It has been reported that among patients with coronary artery disease, approximately 35 and 15% of patients were either diabetic or pre-diabetic, respectively [7]. In addition, individuals with high postprandial glucose (PPG) levels have a higher risk of developing cardiovascular diseases and all-cause mortality than individuals with high fasting plasma glucose (FPG) levels [4,5,6,8]. Therefore, it is important to control PPG levels as well as FPG levels for the appropriate management of diabetic complications.

*Schisandra chinensis*, a berry referred to as ‘Omija’ in Korea, possesses five distinct flavors (salty, bitter, sour, pungent, and sweet), and has been used as a medicinal plant in Korea, China, Japan, and many other countries. It has been reported to have various physiological activities such as immunomodulation, anticancer, anti-tumor, central nervous system enhancement, blood circulation improvement, liver function recovery, and antidiabetic effects [9,10,11,12,13]. Omija contains fatty acids such as palmitic acid and stearic acid, as well as organic acids such as [14] schizandrin, ethamigrenal, and gomisin [15]. Previous studies showed that schisandra polysaccharide significantly alleviates the symptoms in diabetic mice (such as weight loss, polydipsia, polyuria, and hyperglycemia), promotes the synthesis of hepatic glycogen, inhibits the decomposition of hepatic glycogen, and improves the lipid metabolism disorders caused by diabetes [16,17,18]. The results of another study with acidic polysaccharide from *Schisandra chinensis* (SCAP) also showed that SCAP could significantly reduce the levels of FPG, serum total cholesterol (TC), triglyceride (TG), LDL-cholesterol, and malondialdehyde (MDA), as well as elevate the levels of insulin and high-density lipoprotein (HDL)-cholesterol, enhance superoxide dismutase (SOD) activity, and improve the pathological changes in islets in T2D rats. SCAP treatment has also been reported to prevent drastic loss in body weight in T2D rats. The available data indicate that SCAP imparts a good therapeutic effect in diabetic rats [19]. Although Omija has been used as a traditional and medicinal plant, there exists only little information on its effect on glycemic control in humans, apart from its high α-amylase and α-glucosidase inhibitory activity, and postprandial plasma glucose-lowering effect both in vitro and in vivo [20].

Dietary soybean is beneficial to health because of its high polyunsaturated fat, fiber, vitamin, and mineral content along with a low saturated fat content [21]. Previous studies showed the protective effects of soy intake on cancer [22,23], cardiovascular disease [24], serum lipids [25], blood pressure [26], and endothelial function [27]. The inverse association between soybean product intake and T2DM has been reported among women in two prospective studies [28,29]. However, the definition of T2DM in the two studies was based on self-report. In addition, there has been no study investigating the effects of soybean product intake on the incidence of T2DM, considering both PPG levels and FPG levels.

We confirmed the synergic effects of Omija extract and soybean mixture (OSM) on fasting and post-glucose and insulin levels, as well as insulin resistance both in vitro and in vivo. The OSM-induced antidiabetic effects have been identified to be mediated by an increase in glucose susceptibility with the elevation of GLUT2 expression and phosphorylation of AMPK, Akt, and IRS in the liver [30]. Consequently, we aim to investigate the efficacy and safety of OSM on the improvement in glycemic control in humans.

## 2. Materials and Methods

### 2.1. Design of Experiments

This experiment was designed as a single-center, randomized, double-blind, and parallel trial with two arms, implemented in accordance with the Helsinki Declaration and the provision of Korea Good Clinical Practice (KGCP) between February and October 2021 at Woosuk University Korean Medicine Hospital. The protocol was approved by the Institutional Review Board of the Woosuk University Korean Medicine Hospital (IRB approval No.: WSOH IRB H2012-03-01) and registered at Clinical Research Information Service (CRIS, http://cris.nih.go.kr (accessed on 18 August 2021), clinical trial No.: KCT0006613). Potential participants were recruited through several ways (banner, newspaper, Woosuk University Korean Medicine Hospital web page, etc.). People who were directly correlated with the researcher of this trial were excluded from the study. A total of 80 participants were randomly allocated to either placebo or test and each given the product at every visit,. They underwent three visits for the confirmation of changes by the administration of the product every 6 weeks (visit 1: week 0, visit 2: week 6, and visit 3: week 12). In particular, the screening test was performed within 2 weeks before visit 1 (week 0), considering the period to minimize the burden from researchers and the effect of changes in the factors influencing glucose levels, such as diet.

### 2.2. Registration and Randomization of Participants

The volunteers signed an informed consent document (ICD) after getting fully well-acquainted with the entire study process, the aim of the study, and the expected results. The subjects who met the following inclusion and exclusion criteria were registered on visit 1 (week 0) and a total of 80 subjects were selected.

Inclusion criteria:

(1) Between 19 and 70 years old on screening test; (2) Subjects without any history of healing diabetes within 3 months before screening test and 100 mg/dL ≤ FPG < 140 mg/dL on screening test; (3) Providing a written consent after education about the study’s aims and goals.

Exclusion criteria:

(1) HbA1c ≥ 6.5% on screening test; (2) Body Mass Index (BMI) < 18.5 kg/m^2^ or 35 kg/m^2^ ≤ BMI on screening test; (3) Participants with a clinically significant disease requiring treatment (i.e., acute or chronic cardiovascular system disease, endocrine system disease, immune system disease, respiratory system disease, kidney and urinary system disease, neuropsychiatric system disease, musculoskeletal inflammation, inflammatory disease, blood and tumor disease, gastrointestinal disease, etc.); (4) Participants who have taken medicine and health-functional food related with glucose within 3 months prior to screening test; (5) Participants with hypersensitivity to the study product or any ingredients in the study product; (6) Participants who have taken antipsychotic medication within 3 months prior to screening test; (7) History of medicine or alcohol abuse; (8) Those who took part in other clinical trials within 3 months before screening; (9) Laboratory tests with the following results: aspartate transaminase (AST), alanine transaminase (ALT) > reference range 3 times upper limit/serum creatinine > 2.0 mg/dL; (10) Pregnancy or breast feeding; (11) Those who were not in agreement with the implementation of appropriate contraception of a childbearing woman; (12) Principal investigator judged inappropriate for participation in study because of laboratory test results, etc.

The registered 80 subjects were assigned to the OSM or placebo group (equally 1:1 ratio) after visit 1 based on the code generated by the block randomization method performed using SAS^®^ system version 9.4 (SAS Institute, Cary, NC, USA). The randomized allocation was conducted to avoid potential bias and balance demographic and baseline characteristics between the groups. The information of the code was required to be maintained by the client, and researchers or subjects of the study were unaware of it during the experiment.

### 2.3. Intervention

The OSM supplement was prepared by MunGyeong Agricultural technology service center (Mungyeong, Korea). Dried Omija (*Schisandra chinensis*) fruits were extracted with 50% ethanol, filtered, and concentrated. Next, the OSM were freeze-dried to powder that mixed Omija extract and soybean (5:1).

One OSM capsule contained 125 mg OSM, 218 mg crystalline cellulose, each 3.5 mg silicon dioxide and magnesium stearate, and was administered to the OSM group while the placebo capsule contained red food coloring instead of OSM to make it visually identical to the test one. Participants were instructed to take 2 capsules once, 2 times per day after breakfast and dinner (Table 1).

### 2.4. Efficacy Outcomes

The FPG level was assessed before intake of glucose solution and the supplements after over 8 h of fasting at the time of screening, visit 1, and visit 3. Next, a 75 g oral glucose tolerance test (OGTT) was performed according to the protocol at visit 1 (week 0) and visit 3 (week 12). Briefly, participants consumed a 75 g glucose solution and the supplements and blood samples were collected at 0, 30, 60, 90, and 120 min. Insulin levels were estimated using 0-, 30-, 60-, and 90-min blood samples. The AUC (area under the curve) was calculated for plasma glucose and insulin for each subject. The level of insulin-related marker was calculated using the formula: HOMA-IR = [FPG (mg/dL) × FPI (μU/mL)]/405, HOMA-β = 20 × FPI (μU/mL)/{FPG (mg/dL)/18–3.5. The level of glucose-related markers (HbA1c, C-peptide, and fructosamine) and glucose metabolism-related markers (AMPK), as well as lipid profile (TC, TG, HDL-cholesterol, LDL-cholesterol, free fatty acid, Apo A1, and Apo B) were assessed using the 0 min blood sample.

### 2.5. Safety Outcomes

For the safety assessment, adverse events (AE), electrocardiogram, vital signs (pressure and pulse rate), and laboratory tests were performed. Hematologist tests included investigation of the levels of hemoglobin, hematocrit, white blood cell (WBC), red blood cell (RBC), and platelets. The blood chemical tests were implemented to determine AST, ALT, gamma-glutamyl transferase (gamma-GT), albumin, total bilirubin, alkaline phosphatase (ALP), lactate dehydrogenase (LD), total protein, blood urea nitrogen (BUN), creatinine, creatine kinase (CK), and high sensitivity C-reactive protein (hs-CRP) levels. Urinalysis was performed by assessing the pH, nitrite, specific gravity, protein, glucose, ketone body, bilirubin, urobilinogen, WBC, and occult blood in the urine.

### 2.6. Diet and Global Physical Activity Questionnaire (GPAQ)

A dietary survey was performed using 3-day food records (2 weekdays and 1 weekend day) for previous weeks at screening, visit 1, and visit 2. Energy, macronutrients (carbohydrate, protein, and fat), and fiber intake were analyzed using a computer-aided nutritional analysis program (CAN-pro, Korean Nutrition Society, Seoul).

The GPAQ is an international physical activity questionnaire that investigates the physical activity practice according to the type of activity (day, place movement, and leisure activity) and calculates the amount of physical activity through MET values and time indicating intensity. The level of physical activity of the subject is diagnosed and evaluated compared to the recommended level.

### 2.7. Diabetic Family History and Anthropometric Assessments

A diabetic family survey implies the assessment of whether the parents and siblings of the study subjects have been diagnosed with diabetes, and accordingly records the corresponding members if they are identified with the disorder.

The height and weight were measured using the same machine during the test period, and the height (cm) was recorded as measured during the screening visit until the end of the human body application test. The body mass index (kg/m^2^) is the weight (kg) divided by the square of height (m).

### 2.8. Sample Size and Statistical Analysis

The sample size was calculated using the published data from a previous study [19] in G-power 3.1, which was performed by including type 2 diabetic patients who were administered a fruit extract for 2 months. The changes in the levels of the main evaluation biomarker (FPG) after 12 weeks of intake in the OSM and placebo groups were assumed as −20.2 and 7.5 mg/dL, respectively. The number of participants needed in each group was calculated to be 40, for which a power of 80% was obtained with an alpha of 0.05 and a 30% loss to follow-up.

All statistical analyses were performed using SAS^®^ version 9.4 (SAS Institute, Cary, NC, USA). All data are expressed as means ± standard deviations (SD) for continuous variables and as n (%) for categorical variables. The data analysis for efficacy was performed using the full analysis set (FAS) and the analysis for safety was performed using the safety set. We analyzed the data according to the criteria of protocol. Continuous variables collected at baseline or visit 1 were compared between OSM and placebo groups using independent t-tests, while the categorical variables were compared using the Chi-square test (Fisher’s exact test). In the case of comparison within the group, the change at 12 weeks compared to before intake was analyzed using a paired t-test. The changes in the values for 12 weeks between the OSM and placebo groups were analyzed using an independent t-test. Baseline and demographic variables for heterogeneous evaluation items were calibrated with covariates for analysis of covariance (ANCOVA) testing. The significance was statistically significant at the level of *p* < 0.05.

## 3. Results

### 3.1. Characteristics of the Participants

The subjects (n = 80) were randomly allocated to the OSM or placebo group. A total of 63 participants were examined at follow-up according to the criteria of protocol (Figure 1). The demographic characteristics of 63 subjects are shown in Table 2. The ratio of sex, age, height, weight, BMI, vital signs, alcohol consumption, smoking, diabetic history, fasting glucose, and HbA1c was confirmed at the time of the screening test and visit 1. There were significant differences between the groups concerning FPG and HbA1c and we adjusted the baseline of FPG and HbA1c.

### 3.2. Efficacy Outcome

Table 3 and Figure 2 present the changes in the outcome related to glucose metabolism after the intervention. There was a statistical difference in plasma glucose levels at 0, 30, and 60 min within the group and in the OSM group compared to the baseline at the end of the study (*p* = 0.002, 0.023, 0.041). A statistical difference was also observed between the groups when adjusted on baseline FPG, HbA1c (*p* = 0.020, 0.022, 0.002). OSM-administered subjects experienced lower plasma insulin levels and AUC and there was a significant difference in the PPI 60 min and insulin AUC compared with the placebo group even after the adjustment of the baseline FPG and HbA1c (*p* = 0.006, 0.011). Plasma C-peptide and fructosamine levels were significantly reduced within the OSM group (*p* = 0.006, 0.002), but a significant difference was observed between the group on only fructosamine (*p* = 0.033). After taking the supplement, LDL-cholesterol of lipid outcomes reduced significantly within the OSM group (*p* = 0.033), and a significant difference in LDL-cholesterol and free fatty acid was observed when compared to the placebo group (*p* = 0.036, 0.049) (Figure 3). There was no significant difference between the placebo group and the OSM group concerning total cholesterol, triglyceride, HDL-cholesterol, Apo A1, and Apo B.

### 3.3. Diet and Global Physical Activity Questionnaire (GPAQ)

The result of the dietary survey revealed no significant difference between the OSM group and the placebo group in terms of energy, macronutrients (carbohydrate, protein, and fat), and fiber content. Likewise, there was no significant difference in GPAQ (MET value) between the groups (data not shown).

### 3.4. Safety Analysis

A total of two mild adverse events (one headache, one sprain, and one strain of the lumbar spine) occurred in 2 our od 80 participants. However, the adverse events during the study exhibited no clear association with the administration of products, and we observed no significant difference between the groups.

In a laboratory test, the changes in the hemoglobin concentration demonstrated a significant difference between the groups, but that was determined to be within the reference range and had no clinical significance. Other laboratory tests and vital signs were within the reference range (data not shown).

## 4. Discussion

This clinical trial was designed as a placebo-controlled, randomized, and double-blind study to evaluate the efficacy and safety of OSM on the improvement in glycemic control in individuals with hyperglycemia. The administration of OSM for 12 weeks led to positive changes in glucose and insulin-related parameters and no clinically meaningful adverse events were observed, thereby providing evidence of the probable use of OSM as a safe control agent.

We confirmed the synergic effects of OSM in screening tests [30]. In other words, OSM exhibited much better inhibitory effects on α-glucosidase, α-amylase activities, and DPPH scavenging compared with Omija extract or soybean powder. When OGTT was performed to compare the anti-glucose tolerance effect of samples for three types of extraction solvents of Omija, significant differences were observed in all the Omija ethanol extract groups compared to the control group; moreover, glucose levels in the Omija fruit 50% ethanol extract group were observed to be significantly lower. When mixtures based on the diverse ratio of Omija fruit 50% extract and soybean were orally administrated for OGTT, the glucose levels in all the OSM groups were significantly lower compared to the control group. In particular, the O5:S1 (Omija fruit 50% ethanol extract 5: soybean 1) group showed strong effects compared to the positive control group. Based on this result, three doses of OSM were administrated in db/db mice and the effects were compared with the control group. The low-dose group (OSM 10 mg/kg) showed low correlative effects, whereas the medium-dose group (OSM 30 mg/kg), high-dose group (OSM 100 mg/kg), and positive control group were found to be significantly rescued. According to this result, the daily intake of OSM (100 mg/kg) for the human test was set at 0.5 g/day.

Type 2 diabetes refers to a condition in which glucose is not used in fat, muscles, and liver tissues due to insulin resistance, and blood glucose concentration remains high, thereby leading to an increase in insulin secretion to maintain normal plasma glucose levels. This study was conducted on subjects with impaired fasting glucose and early type 2 diabetes, and it was expected that OSM would lower the blood glucose level and reduce insulin secretion. In this study, a significant reduction in FPG and PPG at 30 and 60 min was observed in the OSM group and there were significant differences between the OSM group and the placebo group. The intake of OSM demonstrated a significant difference in postprandial insulin levels at 60 min between the two groups and decreased the area under the plasma insulin curve (insulin AUC) compared to the control. A previous study has also demonstrated that oral administration of OSM reduced PPG at 30 and 60 min in ICR mice [30], thereby indicating that OSM intake inhibits the activity of alpha-glucosidase. Omija fruit water extract has been reported to improve postprandial glucose at 30 min in SD rats after glucose solution loading [20]. In previous studies, a significant decrease in FPG was observed in patients with type 2 diabetes in a meta-analysis [31]. The alpha-glucosidase inhibitors have been reported to inhibit the decomposition of disaccharides into monosaccharides in the small intestine, delaying digestion and absorption of glucose, thereby leading to a reduction in the sudden rise in postprandial glucose levels following the excessive secretion of insulin [32,33].

In the case of diabetic patients, death from cardiovascular disease is two to four times higher than that of non-diabetics, so dyslipidemia is an active treatment target in diabetic patients [34]. In the UK Prospective Diabetes Study (UKPDS), LDL-cholesterol was analyzed as the most potent predictor of coronary artery disease incidence in type 2 diabetes patients, with an increase in the risk of coronary artery disease by about 60% each time LDL-cholesterol increases by 39 mg/dL [35]. It was found that there was little change in blood LDL-cholesterol concentration in the control group after ingestion of OSM for 12 weeks, while a significant decrease has been reported in the test group. Accordingly, it is believed that the risk of coronary artery disease would decrease with a decrease in the blood LDL-cholesterol levels due to OSM intake. This hypothesis is similar to the results of a study confirming the decrease in blood LDL-cholesterol concentration with diabetic patients after the intake of supplements [36].

Fructosamine is a compound that is formed when glucose combines with a blood protein and has a half-life of approximately 20 days. Therefore, the serum fructosamine concentration reflects relatively recent 2–3-week changes in blood glucose levels [37]. It is useful as an indicator for the long-term treatment of diabetes because it does undergo considerable temporary changes due to meals or other factors [38]. We confirmed that the fructosamine levels in the OSM group improved significantly after the intake of OSM and that there were differences in the level of the compound between the OSM and placebo group similar to the reported case with T2DM patients [37].

## 5. Conclusions

This randomized placebo-controlled clinical trial indicates that the intervention with OSM for 12 weeks led to a significant improvement in the glycemic control in subjects with hyperglycemia.

Based on the results of this study, we intend to develop OSM as a dietary supplement with functional claims in Korea.

## Figures and Tables

**Figure 1 nutrients-14-03159-f001:**
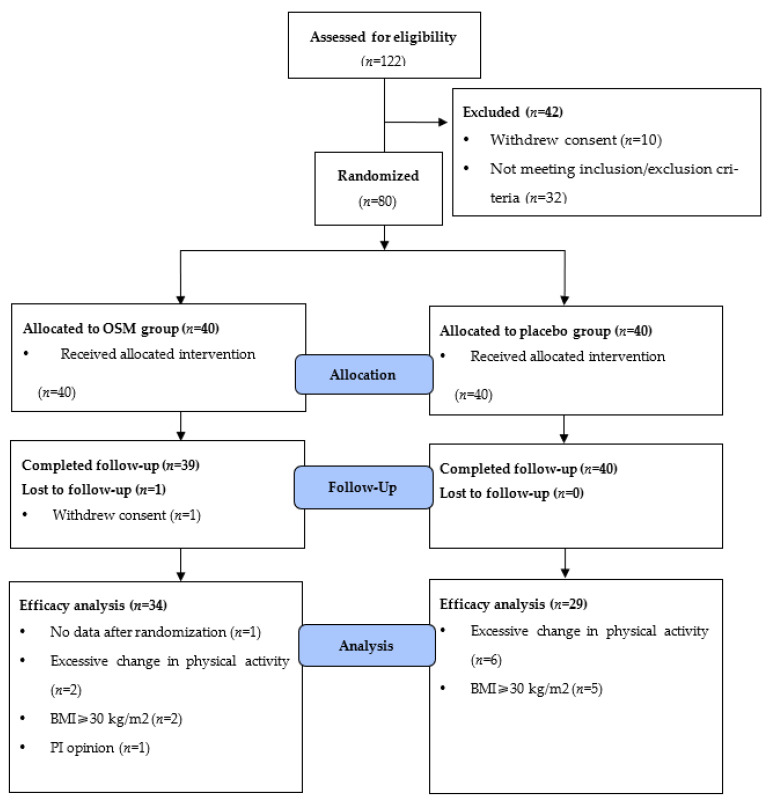
Flow-chart of the participants.

**Figure 2 nutrients-14-03159-f002:**
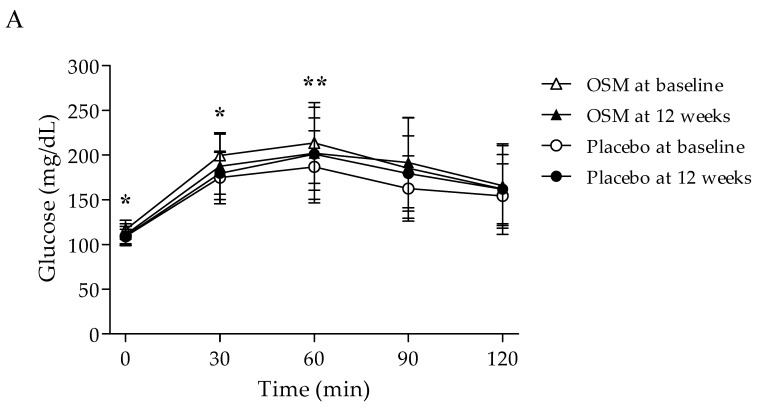
Effects of OSM on plasma glucose and insulin levels during the oral glucose tolerance test (OGTT) after taking the supplement for 12 weeks. (**A**) Plasma glucose level; (**B**) Plasma insulin level. Subjects in the OSM group were administered OSM at 500 mg/day for 12 weeks. Data are expressed as means ± SD. Between-group differences were assessed using ANCOVA (adjusted on the baseline of FPG, HbA1c). * *p* < 0.05, ** *p* < 0.01 vs. placebo group.

**Figure 3 nutrients-14-03159-f003:**
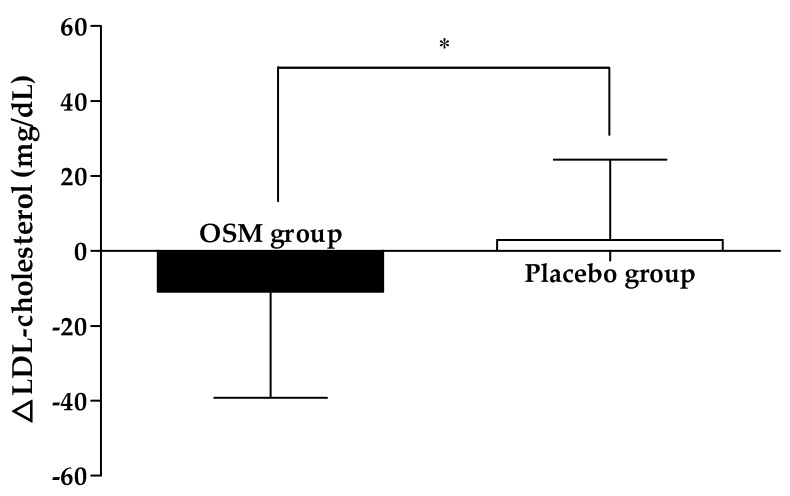
Effects of OSM on changes in plasma LDL-cholesterol and free fatty acid levels from baseline after taking the supplement for 12 weeks. Subjects in OSM groups were administered OSM at 500 mg/day for 12 weeks. Data are expressed as means ± SD. Between-group differences were assessed using ANCOVA (adjusted on the baseline of FPG, HbA1c). * *p* < 0.05 vs. placebo group.

**Table 1 nutrients-14-03159-t001:** Composition of the supplements.

Component	OSM Supplement (%)	Placebo Supplement (%)
OSM (Omija extract and soybean mixture)	35.7	0.0
Crystalline cellulose	62.3	96.8
Red food coloring	0.0	1.2
Silicon dioxide	1.0	1.0
Magnesium stearate	1.0	1.0
Total	100.0	100.0

**Table 2 nutrients-14-03159-t002:** Demographic characteristics of the participants.

	OSM Group (*n* = 34)	Placebo Group (*n* = 29)	Total (*n* = 63)	*p*-Value ^(1)^
Sex (M/F)	18/16	13/16	31/32	0.521 ^(^^2)^
Age (years)	50.82± 9.55	49.38 ± 10.74	50.16 ± 10.06	0.574
Height (cm)	165.26 ± 11.08	163.14 ± 8.26	164.29 ± 9.86	0.398
Weight (kg)	68.74 ± 13.10	65.72 ± 11.20	67.35 ± 12.26	0.334
BMI (kg/m^2^)	24.96 ± 2.43	24.54 ± 2.53	24.77 ± 2.46	0.507
SBP (mmHg)	127.09 ± 10.78	126.28 ± 10.34	126.71 ± 10.50	0.762
DBP (mmHg)	79.50 ± 8.57	79.31 ± 9.66	79.41 ± 9.01	0.935
Pulse (beats/minute)	77.18 ± 9.66	80.34 ± 10.27	78.63 ± 9.99	0.212
FPG (mg/dL)	116.62 ± 10.55	109.00 ± 8.19	113.11 ± 10.20	0.003 **
HbA1c (%)	5.83 ± 0.29	5.67 ± 0.34	5.76 ± 0.32	0.048 *
Alcohol (n, %)	13 (38.24)	12 (41.38)	25 (39.68)	0.799 ^(^^2)^
Alcohol (units/week)	10.54± 5.17	14.53± 5.30	12.45± 5.51	0.070
Smoking (n, %)	5 (14.71)	3 (10.34)	8 (12.70)	0.604 ^(^^2)^
Smoking (cigarette/day)	10.60± 2.61	10.67± 4.04	10.63± 2.92	0.978
Diabetic family history (*n*, %)	13 (38.24)	8 (27.59)	21 (33.3)	0.372 ^(^^2)^

Values are presented as mean ± SD or number (%), * *p* < 0.05, ** *p* < 0.01, ^(1)^ Analyzed by independent t-tests between the groups ^(2)^ Analyzed by chi-square tests between the groups.

**Table 3 nutrients-14-03159-t003:** Changes in outcomes related to glucose metabolism after intake of OSM or placebo supplements.

		OSM Group (n = 34)				Placebo Group (n = 29)				*p*-Value ^(2)^	*Adj.* *p*-Value ^(3)^
		Baseline	12 Weeks	Change in the Value	*p*-Value ^(1)^	Baseline	12 Weeks	Change in the Value	*p*-Value ^(1)^		
FPG (mg/dL)	0 min	116.62 ± 10.55	111.00 ± 12.07	−5.62 ± 9.73	0.002	109.00 ± 8.19	109.66 ± 10.38	0.66 ± 12.25	0.775	0.027 *	0.020 *
PPG (mg/dL)	30 min	199.56 ± 24.04	187.74 ± 37.42	−11.82 ± 28.98	0.023	174.93 ± 29.37	179.59 ± 23.47	4.66 ± 26.36	0.350	0.022 *	0.022 *
	60 min	213.62 ± 45.16	202.15 ± 51.56	−11.47 ± 31.36	0.041	186.76 ± 40.25	201.14 ± 40.45	14.38 ± 32.84	0.026	0.002 **	0.002 **
	90 min	185.41 ± 56.13	191.62 ± 50.41	6.21 ± 38.68	0.356	162.59 ± 36.46	179.48 ± 42.14	16.90 ± 30.08	0.005	0.232	0.232
	120 min	162.00 ± 50.61	165.82 ± 44.72	3.82 ± 38.22	0.564	154.41 ± 35.99	161.90 ± 38.72	7.48 ± 31.91	0.217	0.685	0.685
FPI (μU/mL)	0 min	10.08 ± 10.07	7.17 ± 4.65	−2.91 ± 8.45	0.053	8.85 ± 6.72	7.13 ± 3.74	−1.72 ± 5.99	0.134	0.527	0.527
PPI (μU/mL)	30 min	44.85 ± 34.93	40.38 ± 25.53	−4.46 ± 20.94	0.223	44.38 ± 32.10	45.17 ± 32.49	0.78 ± 19.44	0.830	0.310	0.310
	60 min	59.06 ± 34.03	42.77 ± 19.53	−16.29 ± 27.34	0.002	54.60 ± 35.22	59.42 ± 40.71	4.82 ± 32.01	0.424	0.006 **	0.006 **
	90 min	53.71 ± 33.93	51.39 ± 30.60	−2.32 ± 27.10	0.621	45.00 ± 27.19	56.26 ± 38.59	11.26 ± 31.00	0.061	0.068	0.068
	120 min	45.60 ± 34.53	44.18 ± 30.37	−1.41 ± 20.84	0.695	47.97 ± 32.78	58.02 ± 53.02	10.05 ± 37.13	0.156	0.147	0.129
AUC	Glucose (mg × min/dL)	8168.82 ± 3679.67	8281.76 ± 3570.23	112.94 ± 2608.49	0.802	6614.48 ± 2944.75	7720.86 ± 3043.42	1106.38 ± 2217.94	0.012	0.112	0.112
	Insulin (μU × min/mL)	4357.84 ± 2200.15	3948.49 ± 1996.45	−409.35 ± 1374.66	0.092	4110.19 ± 2524.17	4947.18 ± 3684.09	836.99 ± 2345.70	0.065	0.016 *	0.011 *
HOMA-IR		3.01 ± 3.26	2.00 ± 1.43	−1.02 ± 2.78	0.041	2.43 ± 2.04	1.96 ± 1.19	−0.47 ± 2.00	0.218	0.381	0.381
HOMA-β		64.56 ± 54.45	55.01 ± 32.55	−9.55 ± 45.11	0.226	68.69 ± 45.12	55.86 ± 27.53	−12.83 ± 32.01	0.040	0.744	0.744
HbA1c (%)		5.83 ± 0.29	5.92 ± 0.24	0.09 ± 0.16	0.002	5.67 ± 0.34	5.81 ± 0.31	0.14 ± 0.17	<.0001	0.227	0.195
C-peptide (ng/mL)		2.86 ± 1.43	2.30 ± 0.93	−0.56 ± 1.11	0.006	2.68 ± 1.13	2.25 ± 0.82	−0.42 ± 0.96	0.025	0.615	0.615
Fructosamine (μmol/L)		262.88 ± 24.46	254.97 ± 20.90	−7.91 ± 13.78	0.002	254.62 ± 18.93	253.79 ± 15.86	−0.83 ± 11.67	0.706	0.033 *	0.033 *
AMPK (ng/mL)		7.73 ± 6.91	8.08 ± 5.35	0.35 ± 5.61	0.722	7.68 ± 2.98	11.40 ± 8.45	3.72 ± 7.58	0.013	0.047 *	0.047 *

Values are presented as mean ± SD, * *p* < 0.05, ** *p* < 0.01 ^(1)^ Analyzed by paired t-tests between baseline and 12 weeks ^(2)^ Analyzed by independent t-tests between the groups ^(3)^ Analyzed by ANCOVA (adjusted on the baseline of FPG, HbA1c). Abbreviations: FPG, fasting plasma glucose; PPG, postprandial glucose; FPI, fasting plasma insulin; PPI, postprandial insulin; AUC, area under the curve; HOMA-IR, Homeostatic Model Assessment for Insulin Resistance; HOMA-β, Homeostatic Model Assessment for beta-cell function; QUICKI, the quantitative insulin sensitivity check index; AIR, Acute Insulin Response; AMPK, AMP-activated protein kinase.

## Data Availability

The datasets generated and/or analyzed during the current study are available from the corresponding author on reasonable request.

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
