# Peer review of "Efficacy and Safety of Omija (Schisandra chinensis) Extract Mixture on the Improvement of Hyperglycemia: A Randomized, Double-Blind, and Placebo-Controlled Clinical Trial"

_nutrients, 2022, doi:10.3390/nu14153159_

Round 1
Reviewer 1 Report
In this study, Dasom-Kim et al investigated the beneficial effects of Omija extract and soybean mixture (OSM) on hyperglycemia subjects. It is interesting and has some practical significance. However, some concerns should be addressed before acceptance.
1) The authors should do the intention-to-treat analysis, which included data from all participants who underwent randomization.
2) The authors should give the detailed reasons of protocol violation.
3) This research used Omija extract and soybean mixture to intervene hyperglycemia subjects. However, the title didn’t mention the soybean.
4) Figure 1 has some garbled characters which should be corrected.
Author Response
Response to Reviewer 1 Comments
Reviewer 1
Point 1: The authors should do the intention-to-treat analysis, which included data from all participants who underwent randomization.
Response 1: We added information of analysist in author contributions.
Point 2: The authors should give the detailed reasons of protocol violation.
Response 2: We analyzed the data according to the criteria of protocol written the factors to affect glycemic control and please confirm part of efficacy analysis in Figure 1.
Point 3: This research used Omija extract and soybean mixture to intervene hyperglycemia subjects.
However, the title didn’t mention the soybean.
Response 3: Because our main material is Omija to confirm glycemic control with subjects, we would like to focus on Omija than soybean in title. Additionally, I wish you read second paragraph in 4. Discussion again that we explain why soybean was added to Omija extract.
Point 4: Figure 1 has some garbled characters which should be corrected.
Response 4: We modified Figure 1.
Reviewer 2 Report
The authors of this study provide insight into the therapeutic properties of Schisandra chinensis or Omija and soybeans on type 2 diabetes. It is interesting to mention that there are not many scientific studies on this fruit, although there are on soybeans. It also stands out from this work that an omija/soy mixture was made, since it is easier to assess the effects of omija alone according to its study carried out on experimental animals. In addition, it is important to mention that epidemiological studies of the use of Omija in patients with diabetes are scarce and that is why this work provides important knowledge to scale previous results in experimental animals to diabetic patients, although global statistics limit the scope of this work because the number of patients included in the study is limited. My recommendation to authors is that they should use a single abbreviation for type 2 diabetes since they use T2D, T2DM and it is no longer necessary to use the term mellitus according to the American Diabetes Society.
Author Response
Response to Reviewer 2 Comments
Reviewer 2
Point 1: The authors of this study provide insight into the therapeutic properties of Schisandra chinensis or Omija and soybeans on type 2 diabetes. It is interesting to mention that there are not many scientific studies on this fruit, although there are on soybeans. It also stands out from this work that an omija/soy mixture was made, since it is easier to assess the effects of omija alone according to its study carried out on experimental animals. In addition, it is important to mention that epidemiological studies of the use of Omija in patients with diabetes are scarce and that is why this work provides important knowledge to scale previous results in experimental animals to diabetic patients, although global statistics limit the scope of this work because the number of patients included in the study is limited. My recommendation to authors is that they should use a single abbreviation for type 2 diabetes since they use T2D, T2DM and it is no longer necessary to use the term mellitus according to the American Diabetes Society.
Response 1: We modified ‘Type 2 diabetes mellitus’ to ‘T2DM’.
Round 2
Reviewer 1 Report
I have no more questions.